# Physical Activity and Habitus: Parental Support or Peer Support?

**DOI:** 10.3390/ijerph20032180

**Published:** 2023-01-25

**Authors:** Long Niu, Jing Xu, Yiting E

**Affiliations:** Department of Sociology, Xi’an Jiaotong University, Xi’an 710049, China

**Keywords:** social support, parental support, peer support, physical activity, activity habitus

## Abstract

Social support can affect an individual’s physical activity and activity habitus. This study aims to explore: (1) the correlations between physical activity and parental/peer support among Chinese college students; (2) the differences between the effects of parental/peer support on college students’ physical activity. To achieve these aims, we conducted a cross-sectional study from September to December 2021, recruiting 1005 students (479 male respondents) from seven comprehensive universities in western China. Through OLS linear regression and quantile regression, we empirically testify that both parental support and peer support have significant influence on physical activity and activity habitus of Chinese college students, and peer support has greater impact. We also examine such effects of both types of support by grouping the samples into two groups on the bases of their existing physical activity habitus. We posit that as the most intimate interpersonal relationships, parental support and peer support play a key role in college students’ physical activity and activity habitus, and peer support has greater impact as they step into college. Our study provides insights into the factors of physical activity and activity habitus and we propose that attention regarding the impact of social support should be paid when we are trying to promote the physical activity of college students.

## 1. Introduction

Physical exercise plays an important role in improving the physical quality of college students and cultivating high-quality youth talents. It is prerequisite for the realization of the goal of ”Healthy China”. Regular physical exercise can reduce negative emotions, promote physical health, enhance physical fitness, and improve physical and mental health [1,2,3]. However, according to a general survey, the physical condition of Chinese college students has shown a downward trend in some aspects in recent years [4]. Specifically, college students show no physical exercise plan, low exercise frequency, and other problems. It can be seen that the overall situation of physical exercise of Chinese college students is not optimistic at present, their consciousness of independent exercise is weak, and it is difficult for them to form a good exercise habit. In order to further understand the current problems and reverse the situation of insufficient physical exercise among Chinese college students, it is necessary to explore the factors that affect their physical exercise.

There are many factors that influence individual physical activity, such as exercising infrastructure, environment and overall culture for physical activity and sports [5,6]. Increasingly more researchers are investigating the influence of social support on physical activity [7,8,9]. Social support is defined as “a network of family, friends, neighbors, and community members who provide psychological, physical, and financial help in times of need” [10]. Studies have found that parental and peer support, two types of the main social support at the interpersonal level, have the greatest impact on individual physical activity [11]. Individuals’ perception of emotional and practical support from family and companion will lead to the participation of physical activity [12]. The influence of both parental support and peer support on university students’ physical activity has been discussed to some extent. However, in the Chinese context, how does social support affect college students’ physical activity behaviors? How do parental support and peer support affect college students’ physical activity, respectively, based on the consideration of China’s particular family culture? Which factor is more influential? Obviously, the existing studies have not answered these questions thoroughly and need further analysis.

On the basis of this research gap, we conducted a cross-sectional survey of seven colleges in western China; we aim to examine the following questions: (1) Are there any correlations between physical activity and parental/peer support among Chinese college students? (2) How do parental support and peer support affect the physical activity of college students?

## 2. Literature Review and Development of Research Hypotheses

### 2.1. Social Support and Physical Activity

In recent years, studies have found that parental support has a significant impact on adolescent physical activity. Specifically, there is a positive correlation between parental behavior (including encouragement, instrumental behavior, and general support) and working habit, and some scholars have explored the influence at adolescents’ physical activity level [13,14,15]. A meta-analysis shows that parents accompanying their children in physical activity has a 17% difference in their physical activity behaviors [16]. Parents’ views on the value of physical activity value can predict their children’s physical activity behavior [17]. Wheeler uses supportive strategies, non-supportive strategies, and practical support, showing that children’s physical activity goals can be fulfilled through parental support [18]. Therefore, parental support affects adolescents’ participation in physical activity in different ways.

The interaction with peer groups is of great importance on the process of individual’s socialization. Charles Curry, an American sociologist, who proposed the concept of the primary group, believed that the peer group, with the characteristics of close face-to-face communication and cooperation, is one of the most important parts of the primary group [19]. Prior research has fully demonstrated the importance of peer groups in the development of adolescents. They insist that their physical activity habitus not only would keep them passionate about physical activity but also would release the negative emotion if their peer groups do so. Thus, the importance of peer groups on physical activity requires more investigation [20].

Existing research shows that both parental support and peer support have a significant impact on adolescents’ physical activity. Keating found that peer support has a more significant impact on adolescent physical activity than parental support [21]. Thus, we propose the first hypothesis.

**Hypothesis** **1.***Both parental support and peer support significantly impact the physical activity of college students, and peer support has greater impact*.

### 2.2. Social Support and Physical Activity Habitus

“Habituation” is considered by Bourdieu as a historical change in the system of dispositions and behaviors [22], a bodily behavior that is formed by the accumulation of historical experiences of an individual’s behavior. In the case of physical exercise habit, the perception, thinking and behavior patterns of previous experiences of exercise are internalized and engraved on the mind and body, thus ensuring the consistency and historical and unchanging character of physical practice, which is a continual process of reinforcement. According to Bourdieu’s “habituation” theory, the formation of individual physical exercise habit depends on the interaction with external fields, and family and school are two important fields for the formation of physical exercise habit among college students. At the same time, the formation of physical activity habit among college students is objectively constrained by the parental support (family) and peer support (school) fields. This constraint on college students’ physical activity habits (and formation), i.e., the basis for practice, is sometimes the result of practice. According to Bourdieu, the formation of such habits is an “enduring system of transferable endowments”, which is internalized and incorporated into oneself in an unconscious manner by the objective conditions of one’s existence and social experiences. Through this enduring endowment system, certain behavioral changes can be modified and developed, showing some continuity and transferability between the process of acquiring certain experiences (e.g., family experiences) and other domains (e.g., school experiences), and finally, a consistent endowment tendency of behavioral styles [23]. 

Physical activity habits are a key factor in lifelong physical activity among college students and a direct driver of their motivation. The acquisition of external social support can cause the physical activity habits of college students to be successfully generated. Accordingly, parental support and peer support have a major influence on the formation of physical activity habits of college students; however, these two types of support are not independent of each other but together have an important stage influence on the formation of college students’ physical activity. According to Bourdieu [24], the complementarity of the two concepts of habituation and field constitutes a “specific field of physical activity”; however, there are different types and forms of fields. Therefore, social support in different fields has significant differences on the formation of physical activity habituation among college students. Parental support (family) and peer support (school) are in different fields, and their influence on the formation of physical activity habits is not constant, but dynamic and intertwined. The dual effect of such dynamic features influences the stage and degree of formation of physical activity habits among college students.

It is important to note that the field is a relatively independent social space with mutual distinguishing signs. According to developmental psychology, the growth stage of individuals until the age of seventeen is identified as the adolescent period. For adolescents in this period, in terms of the field space of interaction, parents are their most important objects of interaction, and they rely more on parental support and assistance; thus, whether individuals like or participate in physical activity is closely related to parental support, and parents’ support is the formation and belief inculcation of physical activity, which helps adolescents to form physical activity habits [14]; therefore, Hypothesis 2 is proposed.

**Hypothesis** **2.**
*Parental support significantly influences the formation of physical activity habits among college students.*


However, for the college student population, their physiology and psychology have gradually tended toward becoming adults, especially after entering the university field. With the enrichment of physical activity content and the deepening of role perception, they will realize the importance of physical activity. Especially after entering the university, they begin to have extensive contact with members outside the family, such as friends, classmates, roommates and other peers, so their physical activity may be influenced by peer support influence. Therefore, Hypothesis 2a is proposed.

**Hypothesis** **2a.**
*Peer support significantly influences the formation of physical activity habituation among college students.*


In addition, Chinese college students’ long-term study and life are interactions with their peers, and their physical activity will increase with such interactions. Thus, it can be seen that the effects of parental support and peer support on Chinese college students’ physical activity habits may have different utility de-pending on the field they are in and the individual’s developmental stage and this result will shift the formation of their physical activity habit from parental support to peer support. On the basis of this, we present Hypothesis 2b.

**Hypothesis** **2b.**
*Peer support has a greater effect on the formation of physical activity habit among college students than does parental support.*


### 2.3. Social Support, Physical Activity, and Physical Activity Habitus

Much research on physical activity and activity behavior shows that the preference for physical activity correlates with individual’s hobbies and habitus [25,26]. The subjective willingness to physical activity habitus is an irrational factor during the formation. [27]. According to Bourdieu’s theory, behavior is dominated by habitus. It is like an internal “system,” which is based on persistent internalization of various habitus, “just like society writes into the body”, habitus can become the subconscious basis of individual behavior, thought and emotion under any possible circumstances. Individual’s physical activity habitus is the most important body operational logic in the process of their socialization of physical activity. The experience results generated by this operation logic are the critical foundation for the formation of physical activity habitus, as well as the dynamic mechanism for the formation of physical activity habitus of university students. Social support can be a good source of motivation for the formation of university students’ physical activity habitus and has an important impact on their physical activity [28].

In recent years, researchers have found that interpersonal relationships are related to the improvement of individual physical activity behavior. Parental support and peer support can effectively affect the formation of their activity habitus by encouraging individuals to practice physical activity repeatedly. Individuals can generate self-identification in areas of support and inspire individual’s motivation to participate in physical activity habitus [29]. Parents and peers play an irreplaceable role in establishing individual physical activity habitus. Some studies have found that the impact of parental and peer support on individual’s physical activity differs in various growth stages [30]. Therefore, is the physical activity habitus of university students also affected by parental and peer support? Is there any difference between the two kinds of support? Existing studies have not yet answered these questions.

Furthermore, the previous studies have not reached a consistent conclusion on regarding the correlations between social support and subjective individual’s physical activity habitus. On the one hand, it is commonly believed that as for social interaction, social support is the prior standard of subjective experience of individual physical activity and directly influences the participation of individual’s physical activity [31]. On the other hand, the theory of a cognitive emotional system holds that the behavior of individual physical activity is generated through the interaction between individuals and social relations on the basis of subjective experience [32]. Their physical activity is the dynamic development result of “cognition and experience” based on interpersonal relationships (parents, peers) and activity habitus [33]. It was found that parental and peer support is the antecedent of the formation of university physical activity habitus, and activity habitus is the ultimate basis for their lifelong physical activity. What role do parental support and peer support play in the formation of university students’ physical activity habitus? Are the two related? Moreover, will parental support and peer support have different effects on university students’ physical activity because they have no habitus of physical activity? To answer this question, we propose Hypothesis 3a and Hypothesis 3b.

**Hypothesis** **3a.**
*For those individuals who have not formed physical activity habitus, parental support significantly affects university students’ physical activity.*


**Hypothesis** **3b.**
*For individuals who have formed physical activity, peer support significantly affects university students’ physical activity.*


## 3. Methods

### 3.1. Data

In this study, we selected our respondents from seven general universities in Shan’xi and Gansu provinces, which are two of the most representative universities in western China. We used a multi-staged, randomized sampling method to collect our respondents from Shaanxi Normal University, Tianshui Normal University, Xi’an University of Electronic Science and Technology, Xi’an University of Technology, Xi’an Jiaotong University, Xi’an University of Technology, and Xi’an University of Science and Technology, which are the chief universities in Xi’an city. Our research was conducted during 13th September to 12th December in 2021, and the sample size was 1044. The steps of data collection were as follows. Firstly, we randomly selected the classes from the seven universities. Secondly, the questionnaires were assigned to the selected classes, and all the respondents in the classes were asked to fill in the questionnaire guided by well-trained investigators so as to ensure the validity of questionnaire. It should be noted that the investigators’ guidance has not been involved with the discussion of questionnaire contents, only providing some instructions when filling out the questionnaire. Among all respondents, 39 students were excluded due to the missing values on some core variables. Ultimately, the valid sample size was 1005, and 479 of them were male students. 

#### 3.1.1. Physical Activity of University Students

The physical activity of University students was measured with the International Physical Activity Scale (Short Volume). In the original questionnaire, the respondents were asked about the frequency and duration of low, moderate, and high intensity physical activity per week. To facilitate students’ understanding of physical activity, the listed physical activity items were partially modified and explained under the premise that the meaning of the original questionnaire remained unchanged. In strict accordance with the principles of abnormal values and truncation of the physical activity scale, and the weight of different physical activity, the physical activity energy consumption (MET) of each student was calculated. The low intensity MET was assigned 3.3, the medium intensity MET was assigned 4.0, and the high intensity MET was assigned 8.0 [34]. MET value was used to reflect the physical activity of university students. In order to make the variable of physical activity energy consumption more consistent with the normal distribution, the natural logarithm was adopted to construct a continuous variable consistent with the normal distribution. Table 1 shows the classification standard of physical activity intensity.

#### 3.1.2. Physical Activity Habitus

There is no unified quantitative standard on how to measure whether an individual has formed a habitus of physical activity. The existing measurement can be roughly divided into three types. The first method is centered around “whether students have activity in the specified time and place”. The second method focuses on whether they participate in sports activity more than three times per week. Thirdly, people believe that physical activity is performed at least three times a week and no less than 30 min each, and the body should be loaded with more than moderate intensity of exercise (heart rate faster than 110 times/min) and keep that tempo for one year. This paper selects the third method to ask university students in the questionnaire whether they have formed physical activity habitus and generalizes them into two types (0 = no physical activity habitus, 1 = physical activity habitus already exists) to obtain the variables of university students’ physical activity habitus. In addition, physical activity habitus was also used as a control variable in this study.

#### 3.1.3. Social Support Questionnaire 

Parental Support Scale and Peer Support Scale designed by Prochaska et al. [35] were used in our study. Considering the actual situation of Chinese college students, the listed items are partially modified and explained under the condition that the original concept of the questionnaire remains unchanged. Each item consists of a four-point Likert scale from 1 “never” to 4 “every day”. Everyone is asked about the perceived support for the physical activity (PA) during the past week to measure social support, family support, and peer support, respectively.

The items of parental support are: (1) parents encourage children to engage in PA or other types of sports, (2) parental involvement in PA or other types of sports with children, (3) parents watch children engaging in PA or sports, (4) parents tell children that they are doing well in PA or sports. The Cronbach’s α in our study was 0.909, showing high reliability of the scale.

The items of peer support are: (1) adolescents encourage their friends to engage in PA or other kinds of sports, (2) friends encourage adolescent to engage in PA or other kinds of sports, (3) friends engage in PA or other kinds of sports with adolescents, (4) friends tell adolescents that they are doing well in PA or other kinds of sports. The Cronbach’s α in the present study was satisfactory for peer support (0.923).

#### 3.1.4. Control Variable

In this study, we control sociodemograhic variables, individuals’ physical health, parents’ education level, and individuals’ socioeconomic characteristics. Among them are gender (0 = male, 1 = female), students’ grade (1 = freshman, 5 = graduate or above, measured as a continuous variable), physical health (1 = very poor, 2 = average, 3 = very good), and physical activity habitus (0 = no physical activity habitus, 1 = physical activity habitus). Parental education level was measured on the basis of the higher education level of both parents, recoded as the years of education (primary school and below = 6, junior high school = 9, senior high school = 12, undergraduate = 16, postgraduate and above = 19). Family economic status was measured as a categorical variable, where 1 = lower level, 2 = middle level, and 3 = upper level. Table 2 displays the descriptive statistics of all variables.

### 3.2. Analysis Strategy

The empirical analysis consists of four parts. First, in order to verify Hypothesis 1, ordinary least squares (OLS) linear regression model was used to estimate the influence of both parental support and peer support on the university students’ physical activity intensity. In addition, we compared the coefficients of both types of support to specify whether parental support or peer support has greater influence on students’ physical activity. OLS reflects the influence of both parental support and peer support on university students’ average physical activity intensity, without explaining the impact on different quantiles of university students’ physical activity intensity. The influence of parental support and peer support for college students’ physical activity may vary with the intensity of activity. Consequently, the quantile regression model was applied to select the estimation results of the 0.10, 0.25, 0.5, and 0.75 quantiles representing the conditional distribution of physical activity intensity and to compare them with the estimation results of ordinary least squares OLS. Secondly, as college students’ physical activity habitus variables are dichotomous variables, we also use binary logistic regression to investigate the impact of parental support and peer support on students’ physical activity habitus. Thirdly, to further specify the influence of both types of support for different habitus, we also used OLS regression model to test the influence on habitual exercise groups and their counterparts. Finally, the total time of physical activity of college students was selected instead of the energy expenditure of physical activity, and a robustness analysis of physical activity habitus was carried out as the dependent variable. All analyses were performed using the stata16.0 software.

## 4. Results

### 4.1. Physical Activity Status of Chinese University Students

Table 3 shows the physical activity of university students with different intensities every week. According to the distribution in Table 3, 733 university students achieved high intensity physical activity every week in the survey sample, and 272 did not achieve high intensity physical activity. Among them, the number of people carrying out high intensity for 2 days per week was the largest, accounting for 20.79% of the total sample, and the number of people carrying out high intensity for 7 days per week was the smallest, accounting for 2.20% of the total sample; it was found that 654 university students had moderate physical activity every week and 351 did not. Among them, the number of people who carried out moderate intensity one day per week was the largest, accounting for 22.13% of the total sample, and the number of people who carried out the moderate intensity seven days per week was the smallest, accounting for 2.01% of the total sample; additionally, 972 university students had low intensity physical activity every week, 33 had no low intensity physical activity, and the largest number of students had low intensity physical activity 7 days per week, accounting for 72.99% of the total sample.

### 4.2. Correlation Analysis

Table 4 shows the Pearson’s correlation analysis results among physical activity energy consumption physical activity habitus, parental support, and peer support. There is a high correlation among university students’ physical activity energy consumption, activity habitus, parental support, and peer support without control variables, and the correlation coefficients are significant. Controlling for other variables, the specific effects of parental support and peer support on college students’ physical activity and activity habitus need to be further tested using regression analysis.

### 4.3. Analysis of the Influence of Parental Support and Peer Support on University Students’ Physical Activity

The OLS and quantile regression results of parental support and peer support on college students’ physical activity energy consumption are shown in Table 5. Although parental support and peer support have a certain degree of influence on university students’ physical activity energy consumption, their influence is different. The results of the model show that parental support and peer support positively affect college students’ physical activity energy consumption. Specifically, for each unit of increase in parental support and peer support, university students’ physical activity energy consumption increases by 1.007 (exp (0.007) = 1.007) and 1.01 (exp (0.010) = 1.010), respectively. According to the coefficient, peer support has greater impact on university students’ physical activity energy consumption than parental support. Thus, our Hypothesis 1 has been testified. 

According to the OLS regression results, parental support and peer support had a significant positive effect on the energy expenditure of physical activity in college students, respectively. However, the results could not reflect the distribution of the influence of various explanatory variables on college students’ physical activity energy consumption. As a result, the bootstrap method was used to conduct quantile regression of the influence of parental support and peer support on university students’ physical activity energy consumption. The quantile regression model can describe the relationships between parental support and peer support and college students’ physical activity energy consumption in a more detailed way. Table 5 shows the regression results of parental support and peer support on university physical activity energy consumption at the 0.10, 0.25, 0.50, and 0.75 quantiles. In order to further observe the change trend of the influence of parental support and peer support on university students’ physical activity energy consumption, this study also shows the quantile regression change trend chart of the influence of different explanatory variables on university students’ physical activity energy consumption, as shown in Figure 1 and Figure 2.

It can be found from Figure 1 that, on the whole, the effect of parental support on physical activity energy consumption of college students is lower in high quantiles than their peer support. Parental support has no significant effect on college students’ physical activity energy consumption at the 0.10 and 0.75 quantile points. There are significant positive effects at the 0.25 and 0.50 quantile points. Specifically, at the 0.25 and 0.50 points, the energy consumption of physical activity of college students will increase by 1.007 (exp (0.007) = 1.007) and 1.005 (exp (0.005) = 1.005) for each unit increase of parental support.

It can be seen in Figure 2 that peer support has a significant positive impact on college students’ physical activity energy consumption at all quantiles, showing an “inverted U-shaped” curve shape. Specifically, below the 0.10, 0.25, 0.50, and 0.75 quantiles, the energy consumption of physical activity of college students increases by 1.010 (exp (0.01) = 1.010), 1.012, 1.009, and 1.007, respectively, for each unit increase of peer support.

Generally speaking, parental support and peer support have a positive impact on college students’ physical activity energy consumption, peer support has a greater impact than parental support, and the impact on different quantiles of university students’ physical activity energy consumption condition distribution is greater than parental support. In other words, if university students’ physical activity were affected by both parental support and peer support, then peer support has greater impact on university students’ physical activity than parental support, so Hypothesis 1 is verified. 

### 4.4. The Influence of Parental Support and Peer Support on College Students’ Physical Activity Habitus

As Table 6 shows, we further analyzed whether that parental support and peer support had an effect on college students’ physical activity habitus and which support had a greater impact. Model 1 shows that gender, grade, and physical health status significantly affects college students’ physical activity habitus. However, father’s education level and family economic status have negative effects on college students’ physical activity habitus. Based on Model 1, parents’ support was added to formulate Model 2. The results showed that, when controlling other variables and adding parents’ support alone, college students’ physical activity habitus will increase by 4.3% for each one unit increase of parental support. Model 3 adds peer support based on Model 1. The results show that, when controlling for other variables and adding peer support alone, the occurrence ratio of college students’ physical activity habitus will increase by 5.1% for each one unit increase of peer support. 

Considering that both parental support and peer support have significant positive impact on college students’ physical activity habitus, a multicollinearity test was conducted before Model 4 was established; all the variance inflation factors were 1.39, indicating that there is no multicollinearity problem in Model 4. The results of adding parental support and peer support in Model 4 show that the influence ratio of parental support and peer support on the physical activity habitus of college students has decreased, but it is still significant. Specifically, under the control of other variables, the odds ratio of parental support and peer support to college students’ physical activity habitus increased by 2.4% and 4.0%, respectively. It can be seen from the impact ratio that peer support has greater impact on college students’ physical activity habitus than does parental support. Hence, Model 4 strongly proves that college students’ physical activity habitus is more affected by peer support; thus, Hypothesis 2, 2a, and 2b are verified.

### 4.5. Sub-Sample Regression Results and Analysis

This paper further analyzed the impact of parental and peer support on college students’ physical activity energy consumption among different physical activity habitus by sub-sample regression according to whether individuals have formed physical activity habitus. The results are shown in Table 7. The sub-sample regression indicates that parental support and peer support have different effects on the physical activity energy consumption of college students with or without physical activity habitus. Specifically, Model 1–Model 3 in Table 3 shows that parental support and peer support significantly affect college students’ physical activity energy consumption in the sample who have not formed physical activity habitus, but in the light of coefficient comparison, there was no statistically significant difference between the two, and hypothesis 3a has not been verified; Model 4–Model 6 shows that, for college students who have been accustomed to physical activity(with physical activity habitus), their physical activity is not affected by parental support, while peer support significantly affects college students’ physical activity. Hypothesis 3b has been testified.

### 4.6. Robustness Analysis

In order to provide more reliable research results, our study measured the effects of parental support and peer support on college students’ physical activity by college students’ physical activity energy consumption in the benchmark regression. We found that parental support and peer support had a significant impact on college students’ physical activity and physical activity habitus; peer support had a greater impact. To test the robustness of this conclusion, this section used more indicators to perform further empirical analysis on college students’ physical activity. This paper measured college students’ physical activity by the energy consumption of physical activity. Since energy consumption of physical activity may also vary with the time of physical activity, the total time of low, medium, and high intensity physical activity of college students in the questionnaire of the International Physical Strength Scale (Short Volume) was replaced by the energy consumption of physical activity as a dependent variable for robustness analysis and test. As the physical activity time did not fully conform to the normal distribution, the logarithm of physical activity time was taken in the model for robustness test. 

The results of robustness analysis of physical activity time are shown in Table 8, indicating that (1) parental support significantly affects college physical activity time; (2) peer support significantly affected college physical activity time; (3) compared with parental support, peer support has a greater impact on college students’ physical activity time; (4) parental support significantly affects college students’ physical activity habitus; (5) peer support significantly affects college students’ physical activity habitus; (6) compared with parental support, peer support has a greater impact on college students’ physical activity habitus. In general, the results of the robustness analysis were consistent with the findings of parental support and peer support on physical activity and physical activity habitus of college students.

## 5. Discussion

The results show that parental and peer support has significant impact on physical activity and activity habitus, indicating that improving social support of Chinese college students will significantly promote their physical activity and activity habitus. The correlation between parental support and peer support on college students’ physical activity and activity habitus shows a significant positive consistency. Peer support has a greater impact than parental support, especially on college students who have formed physical activity habitus. These results show that peer support is the of the most important in the process of college students’ socialization of physical activity. 

It is worth mentioning that parental and peer support have significant impacts on promoting college students’ physical activity. This result is consistent with previous views [36,37,38]. It can be seen that parental support and peer support have an impact on individual physical activity not only in young people, but also in college students. However, in terms of the change of university students’ life style, the influence of parental support and peer support for university students’ physical activity shows a notable difference. Existing research on adolescent physical activity are consistent for college students in the early stage of entering university, but there is a lack of consistent models across stages. Our study shows that for Chinese college students, peer support has a higher impact on their physical activity than parental support.

There is also a positive correlation between parental and peer support on physical activity habitus of Chinese college students, and parental support has a weaker impact than peer support. The possible reasons are closely related to Chinese college students’ leaving home and living in school as well as the changes in their psychological cognition. For Chinese college students, the campus is the main environment for their daily life and study. They spend most of their time with their classmates and roommates. Both physically and mentally, they need to establish close relationships and peer recognition. Therefore, peer support has an important impact on college students’ physical activity behavior, and they want to be recognized or supported by their peers when they leave their families and parents for a long time. After entering the university campus life, the objects of their communication gradually shift to their peers, and it is reasonable that parental influence on university students’ physical activity habitus is weaker than peer support.

As for the effect of parental support and peer support on physical activity of Chinese college students in the sub-sample of whether they have formed physical activity habitus, peer support showed a statistically positive significance. This suggests that for university students, the formation of physical activity habitus is a process from external control to internal control. In the initial stage of forming physical activity habitus, external social support factors play a major role. However, with the formation of habitus of physical activity, external support gradually transformed into internal motivation, forming the endogenous power of self-active physical activity. This endogenous motivation shows obvious differences after college students enter the campus. Parental support is no longer significant for college students with physical activity habitus. After entering college, parents exert limited influence on their physical activity. Nevertheless, as the most sympathetic partner, peers can understand each other’s words and deeds and provide effective help. Therefore, as far as the physical activity of college students are concerned, compared with parental support, peer support has a more significant impact on the physical activity of college students with physical activity habitus. Our results show that Bourdieu’s theory of field and habitus is still applicable to Chinese college students’ physical activity.

## 6. Conclusions

This paper highlights the importance of social support for physical activity and activity habitus of Chinese college students. The main contribution of this study is to explore the impact of parental support and peer support on physical activity and activity habitus of Chinese college students. Additionally, we discuss the differences between parental support and peer support on different exercise habitus to enrich the research conclusions.

The main conclusions of this study are summarized below. First, parental support and peer support significantly affect the physical activity of Chinese college students, and peer support has a greater impact. This shows that improving social support is a feasible way to promote physical activity of Chinese college students. Secondly, parental support and peer support significantly affect the physical activity habitus of Chinese college students, and peer support has a greater impact. More specifically, the impact of parental support on their physical activity habitus has gradually declined since they entered a more independent life on campus, while the influence of peer support has increased. Third, in the sample of Chinese college students without physical activity habitus, parental support and peer support significantly affect college students’ physical activity; In the sample with physical activity habitus, parental support has no significant impact on college students’ physical activity energy consumption, while peer support has a significant positive impact on college physical activity. The result shows that it is very necessary to promote the formation of physical habitus to stimulate the physical activity of Chinese college students.

This study is of practical value for specifying the factors of parental and peer support on the physical activity of Chinese college students. Our findings may serve as a realistic basis for further research on the physical exercise of Chinese college students, thus improving their physical fitness, and achieving the strategic goal of a healthy China as soon as possible. Moreover, our empirical research has provided some meaningful enlightenment for future research. Both parental support and peer support have been proved to significantly affect physical activity and activity habitus of Chinese college students. In this perspective, other factors such as teacher support and school support may also have influence on physical activity and activity habitus of Chinese college students. In the future, researchers may focus on these factors to design a multi-period data tracking study.

This study has some limitations. (a) Due to limitation of data, this paper did not further investigate the difference between the influence of teacher support and school support on college students’ physical activity. (b) In the analysis, there are few samples of college students above senior grade, so the results may be biased against senior grade groups. (c) Our results are based on the self-reported survey of respondents, which means that this paper explains only the subjective factors of physical activity of Chinese college students. Despite these limitations, our study provides a relatively complete framework for exploring the impact of social support factors on individuals’ physical activity and activity habitus. Therefore, our study has important theoretical and empirical significance.

## Figures and Tables

**Figure 1 ijerph-20-02180-f001:**
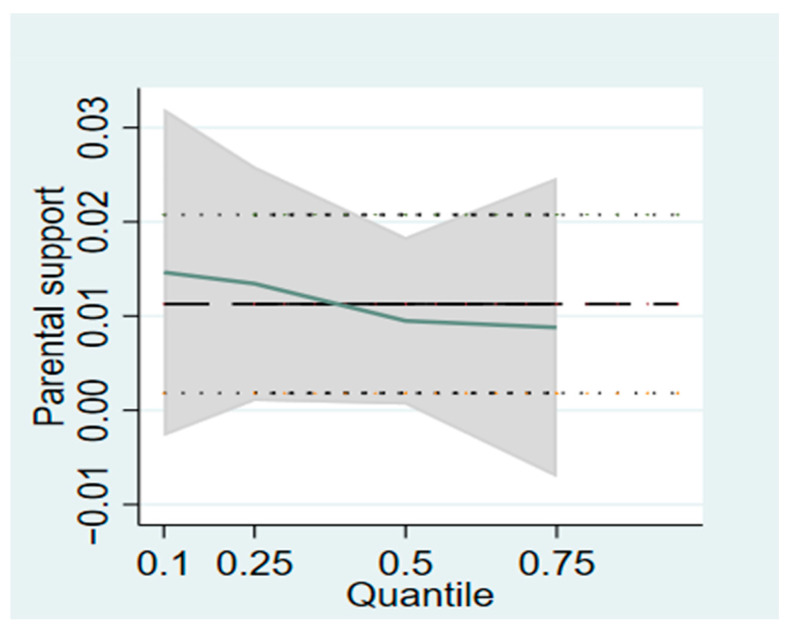
Trend of parental support on physical activity energy expenditure in college students.

**Figure 2 ijerph-20-02180-f002:**
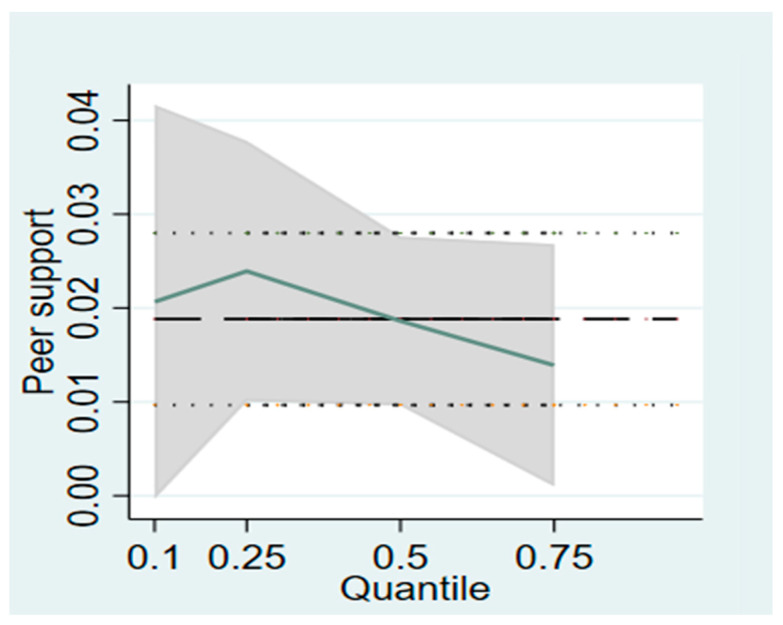
Trend of Influence of Peer Support on Physical Activity Energy Consumption of College Students.

**Table 1 ijerph-20-02180-t001:** Grouping criteria for individuals’ physical activity intensity.

**High**	Inclusions of any one of the following two standards:
	(1) High intensity physical activity ≥ 3 days, and total weekly physical activity level
	1500 MET—min/week
	(2) Total physical activity of three intensities ≥ 7 days and total weekly physical activity level
	≥3000 MET—min/week
**Middle**	Meet any one of the following three standards:
	(1) High intensity implies individual’s physical activity ≥ 20 min per day, and total ≥ 3 days
	(2) Moderate intensity and/or walking ≥ 30 min/day, and total ≥ 5 days
	(3) Total physical activity of 3 intensities ≥ 5 days, and total weekly physical activity level
	≥ 600 MET—min/week
**Low**	Meet any one of the following two standards:
	(1) No physical activity of any intensity was reported
	(2) This physical activity is reported, but they do not meet the above criteria for the medium and high groups

**Table 2 ijerph-20-02180-t002:** Descriptive statistics of the variables.

Variables	N	Mean	Std. Dev.	Type	Min.	Max.
Energy consumption of physical activity (logarithm)	1005	7.891	0.737	Continuous	5.288	9.867
Physical activity habitus	1005	1.306	0.461	Binary	0	1
Parental support	1005	18.554	9.209	Continuous	0	40
Peer support	1005	16.681	9.499	Continuous	0	40
Gender	1005	0.523	0.5	Binary	0	1
Grade	1005	1.741	1.093	Quintile	1	5
Physical health status	1005	2.526	0.619	Triadic	1	3
Parents’ education level	1005	11.608	3.47	Quintile	6	19
Family economic status	1005	2.428	0.746	Triadic	1	5

**Table 3 ijerph-20-02180-t003:** Physical activity of university students.

Activity Intensity	N	No Physical Activity Sample	One Day Week, %	Two Days/ Week, %	Three Days/ Week, %	Four Days/ Week, %	Five Days/ Week, %	Six Days/ Week, %	Seven Days/ Week, %
High strength	733	272	16.19	20.79	16.00	9.10	4.69	4.98	2.20
Moderate strength	654	351	22.13	18.97	11.88	4.31	3.64	3.45	2.01
Low strength	972	33	0.77	2.30	3.64	2.59	9.48	5.08	72.99

**Table 4 ijerph-20-02180-t004:** Correlation analysis.

Variables	Energy Consumption of Physical Activity	Physical Activity Habitus	Parental Support	Peer Support
Energy consumption of physical activity	1.000			
Physical activity habitus	0.391 ***	1.000		
Parental support	0.223 ***	0.197 ***	1.000	
Peer support	0.300 ***	0.267 ***	0.451 ***	1.000

*** *p* < 0.001.

**Table 5 ijerph-20-02180-t005:** Influence of parental and peer support on physical activity energy expenditure (natural logarithm) of college students.

Variables		Energy Consumption of Physical Activity (Logarithm)			
	OLS	0.10	0.25	0.50	0.75
Parental support	0.007 ***	0.007	0.007 **	0.005 *	0.004
	(0.002)	(0.005)	(0.003)	(0.003)	(0.003)
Peer support	0.01 ***	0.01 *	0.012 ***	0.009 ***	0.007 **
	(0.002)	(0.005)	(0.004)	(0.003)	(0.003)
Physical activity habitus	0.434 ***	0.422 ***	0.401 ***	0.361 ***	0.42 ***
	(0.047)	(0.126)	(0.063)	(0.056)	(0.071)
Gender	−0.175 ***	−0.169	−0.208 ***	−0.209 ***	−0.166 ***
	(0.042)	(0.115)	(0.052)	(0.042)	(0.048)
Grade	−0.116 ***	−0.219 ***	−0.167 ***	−0.168 ***	−0.105 ***
	(0.025)	(0.064)	(0.033)	(0.039)	(0.037)
Physical health status	0.143 ***	0.146 **	0.09 **	0.139 ***	0.123 ***
	(0.035)	(0.073)	(0.039)	(0.035)	(0.035)
Parental education level	−0.012 *	0.003	−0.006	−0.006	−0.013
	(0.006)	(0.014)	(0.008)	(0.006)	(0.01)
Family economic status	−0.04	−0.149 **	−0.058	−0.02	0.018
	(0.030)	(0.068)	(0.053)	(0.029)	(0.038)
Cons	6.987 ***	3.606 ***	4.162 ***	4.326 ***	4.727 ***
	(0.155)	(0.305)	(0.231)	(0.222)	(0.306)
R2	0.256	0.119	0.143	0.156	0.158
N	1005	1005	1005	1005	1005

*** *p* < 0.001; ** *p* < 0.01; * *p* < 0.05.

**Table 6 ijerph-20-02180-t006:** Logistic regression models of parental support and peer support on college students’ physical habitus.

Variables	Physical Activity Habitus			
	Model 1	Model 2	Model 3	Model 4
	(Odds)	(Odds)	(Odds)	(Odds)
Parental support		1.043 ***		1.024 **
		(0.008)		(0.009)
Peer support			1.051 ***	1.04 ***
			(0.008)	(0.009)
Gender	0.502 ***	0.467 ***	0.536 ***	0.509 ***
	(0.076)	(0.072)	(0.083)	(0.080)
Grade	0.83 *	0.894	0.885	0.908
	(0.083)	(0.087)	(0.086)	(0.087)
Physical health status	4.131 ***	3.707 ***	3.67 ***	3.528 ***
	(0.645)	(0.583)	(0.578)	(0.558)
Parental education level	0.984	0.971	0.992	0.983
	(0.023)	(0.023)	(0.024)	(0.024)
Family economic status	0.894	0.869	0.888	0.874
	(0.097)	(0.096)	(0.099)	(0.098)
Cons	0.007 ***	0.006 ***	0.005 ***	0.005 ***
	(0.005)	(0.004)	(0.003)	(0.004)
R2	0.119	0.139	0.150	0.155
N	1005	1005	1005	1005

*** *p* < 0.001; ** *p* < 0.01; * *p* < 0.05.

**Table 7 ijerph-20-02180-t007:** Fractional regression results.

Variables	Energy Consumption of Physical Activity (Logarithm)					
	Without Habitus of Physical Activity			With a Habitus of Physical Activity		
	Model 1	Model 2	Model 3	Model 4	Model 5	Model 6
Parental support	0.015 ***		0.011 ***	0.005		0.001
	(0.002)		(0.003)	(0.003)		(0.004)
Peer support		0.015 ***	0.011 ***		0.01 ***	0.01 **
		(0.002)	(0.003)		(0.003)	(0.004)
Gender	−0.198 ***	−0.154 ***	−0.18 ***	−0.194 **	−0.178 **	−0.177 **
	(0.051)	(0.050)	(0.050)	(0.077)	(0.077)	(0.077)
Grade	−0.09 ***	−0.089 ***	−0.08 ***	−0.18 ***	−0.188 ***	−0.188 ***
	(0.030)	(0.030)	(0.030)	(0.049)	(0.049)	(0.049)
Physical health status	0.092 **	0.096 **	0.081 **	0.373 ***	0.36 ***	0.36 ***
	(0.038)	(0.038)	(0.038)	(0.083)	(0.083)	(0.083)
Parental education level	−0.009	0.001	−0.005	−0.027 **	−0.024 **	−0.024 *
	(0.007)	(0.007)	(0.008)	(0.012)	(0.012)	(0.012)
Family economic status	−0.009	−0.001	−0.011	−0.114 **	−0.109 ***	−0.109 **
	(0.036)	(0.037)	(0.036)	(0.053)	(0.053)	(0.054)
Cons	7.456 ***	7.374 ***	7.357 ***	7.67 ***	7.56 ***	7.56 ***
	(0.170)	(0.172)	(0.171)	(0.35)	(0.40)	(0.36)
R2	0.055	0.091	0.093	0.190	0.195	0.210
N	697			308		

*** *p* < 0.001; ** *p* < 0.01; * *p* < 0.05.

**Table 8 ijerph-20-02180-t008:** Results of robustness analysis.

Variables	Physical Activity Time (Logarithm)			Physical Activity Habitus		
	Model 1	Model 2	Model 3	Model 4	Model 5	Model 6
Parental support	0.013 ***		0.007 ***	0.008 ***		0.005 ***
	(0.002)		(0.002)	(0.001)		(0.001)
Peer support		0.015 ***	0.012 ***		0.009 ***	0.008 ***
		(0.002)	(0.002)		(0.001)	(0.001)
Control Variable	Cornered	Cornered	Cornered	Cornered	Cornered	Cornered
Cons	4.80 ***	4.547 ***	4.722 ***	−0.339 ***	−0.396 ***	−0.397 ***
	(0.144)	(0.148)	(0.144)	(0.100)	(0.100)	(0.100)
R2	0.180	0.202	0.208	0.153	0.165	0.171
N	1005	1005	1005	1005	1005	1005

*** *p* < 0.001.

## Data Availability

All data permissions have been obtained and all data in the paper are copyright free.

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
