# Peer review of "Physical Activity and Habitus: Parental Support or Peer Support?"

_ijerph, 2023, doi:10.3390/ijerph20032180_

Round 1
Reviewer 1 Report
Dear Authors,
I am pleased to review an original research article draft "Physical Activity and Habitus: Parental Support or Peer Support?". This piece of work is dedicated to an interesting topic, extends the literature and serves the primary dataset. However, it should be significantly improved. I would invite authors to follow my points, notes and recommendations below:
1. In general, the abstract is logically structured and quite informative. However, it does not seem formulated well. Notably, the research purposes look like research questions (which are different by the way). In order to shape the study purpose/objective, please use ACTION VERBS, but not interrogations.
2. (15) "we also analysis"?
3. (19) "important" should be erased, since it is too subjective and redundant.
4. Also, research originality and implications could be a great extension for the abstract (2-3 short informative sentences, please).
5. (108-110) Sorry, but this is a strange way to represent your hypothesis 2. Unreasonably it is split between manuscript subsections, it raises from the middle of nowhere, unexplained and theoretically unsupported. The best position of the hypothesis is the methods section. But the best way to terminate the introduction is to clearly state research questions AND research objectives, which is missing. The literature review is supposed to shape your THEORETICAL FRAMEWORK well and articulate the constructs.
6. Furthermore, your single hypothesis 2 involves many complex elements. For example, what if Parental support has a significant impact on the formation of college students' physical activity habitus, BUT peer support does not? Or vice versa? Both things potentially can be contradictory...
you should clearly justify and explain WHY parental support and peer support are united together in the hypothesis, OR split it into two hypothesises (like you are done with H3a and H3b): 2a. parental support; 2b. peer support.
7. One factor in the introduction is ignored: infrastructure, environment and overall culture for physical activity and sports. In this regard, I would recommend considering
Glebova, E. and Desbordes, M., 2022. Smart sports in smart cities. Chapter 4 in Buhalis, Taheri, Rahimi, Smart Cities and Tourism: Co-creating experiences, challenges and opportunities: Co-creating experiences, challenges and opportunities. Goodfellow Publishers, Oxford.
https://www.goodfellowpublishers.com/free_files/Chapter%204-1f0758542a746b372f24bac62803c888.pdf
King, A.C., Whitt-Glover, M.C., Marquez, D.X., Buman, M.P., Napolitano, M.A., Jakicic, J., Fulton, J.E. and Tennant, B.L., 2019. Physical Activity Promotion: Highlights from the 2018 Physical Activity Guidelines Advisory Committee Systematic Review. Medicine and science in sports and exercise, 51(6), pp.1340-1353.
8. Due all respect to the authors' choice and creativity, I wonder why section 3 is entitled "measures" instead of methods.
9. The sample should be better explained and justified. Who are they? Why this sample but not another?
10. I would be interested to see the limitations following the conclusion.
Author Response
Dear Reviewer,
On behalf of my co-authors, we thank you very much for giving us an opportunity to revise our manuscript. We appreciate you very much for the positive and constructive comments and suggestions on our manuscript entitled "Physical Activity and Habitus: Parental Support or Peer Support?"(ijerph-2091561). In this revised version, we have addressed the concerns of your suggestions. An item-by-item response to the comments is enclosed, and the revision was marked in RED fonts in the manuscript. We hope that these revisions successfully address your concerns and requirements. Looking forward to hearing from you soon.
Sincerely,
Long Niu
Response to Reviewer 1 Comments
- In general, the abstract is logically structured and quite informative. However, it does not seem formulated well. Notably, the research purposes look like research questions (which are different by the way). In order to shape the study purpose/objective, please use ACTION VERBS, but not interrogations.
- (15) "we also analysis"?
- (19) "important" should be erased, since it is too subjective and redundant.
- Also, research originality and implications could be a great extension for the abstract (2-3 short informative sentences, please).
Response 1-4:We are very sorry to answer these four questions in a unified manner, because I have noticed that your four questions are all amendments to the abstract. In order to solve the above four problems, we have made great changes to the writing of abstract.
- (108-110) Sorry, but this is a strange way to represent your hypothesis 2. Unreasonably it is split between manuscript subsections, it raises from the middle of nowhere, unexplained and theoretically unsupported. The best position of the hypothesis is the methods section. But the best way to terminate the introduction is to clearly state research questions AND research objectives, which is missing. The literature review is supposed to shape your THEORETICAL FRAMEWORK well and articulate the constructs.
Response 5: Thank you very much for your suggestions. We have re-elaborated the literature review in part 2.2 and split the prior hypothesis 2 into seperated part to explore the different effect of parental and peer support on physical activity and activity habitus, which were in LINE 132-133(Hypothesis 2) ; LINE 141-142(Hypothesis 2a); LINE 149-150 (Hypothesis 2b).
- Furthermore, your single hypothesis 2 involves many complex elements. For example, what if Parental support has a significant impact on the formation of college students' physical activity habitus, BUT peer support does not? Or vice versa? Both things potentially can be contradictory...
you should clearly justify and explain WHY parental support and peer support are united together in the hypothesis, OR split it into two hypothesises (like you are done with H3a and H3b): 2a. parental support; 2b. peer support.
Response 6:Thanks for your professional advice. According to your reminding, we have optimized the formulation of hypothesis 2 and divided this part of hypothesis into more specific parts according to the discussion of the research problem
- One factor in the introduction is ignored: infrastructure, environment and overall culture for physical activity and sports. In this regard, I would recommend considering
Glebova, E. and Desbordes, M., 2022. Smart sports in smart cities. Chapter 4 in Buhalis, Taheri, Rahimi, Smart Cities and Tourism: Co-creating experiences, challenges and opportunities: Co-creating experiences, challenges and opportunities. Goodfellow Publishers, Oxford.
https://www.goodfellowpublishers.com/free_files/Chapter%204-1f0758542a746b372f24bac62803c888.pdf
King, A.C., Whitt-Glover, M.C., Marquez, D.X., Buman, M.P., Napolitano, M.A., Jakicic, J., Fulton, J.E. and Tennant, B.L., 2019. Physical Activity Promotion: Highlights from the 2018 Physical Activity Guidelines Advisory Committee Systematic Review. Medicine and science in sports and exercise, 51(6), pp.1340-1353.
Response 7:Thank you very much for your suggestions and specific help to this paper. I have read these literatures and books carefully and quoted them in this paper in LINE 39-40.
- Due all respect to the authors' choice and creativity, I wonder why section 3 is entitled "measures" instead of methods.
Response 8:Thanks for your kind reminds. We have modified it accordingly, replacing “measures” with “methods”.
- The sample should be better explained and justified. Who are they? Why this sample but not another?
Response 9:Thanks for the kind reminds. We justified the reason why we choose these universities as our sampling unit because all of the seven universities we chose are from Shan’xi province and Gansu province, which are two of the most representative provinces in western China.
- I would be interested to see the limitations following the conclusion.
Response 10:Thanks for your reminds, we move the limitations of this study in the last paragraph following the conclusion.

Reviewer 2 Report
I would like to congratulate the authors for the work done and thank them for the invitation to review the manuscript.
I will make some suggestions below
I believe that the first paragraph should specifically specify the problem that the researchers are going to study. However, reading the first paragraph, it is not clear to me.
They should emphasize the problem of physical inactivity in university students,
The entire introductory section must be rewritten except for the last paragraph.
Page 3 Line 100/107. The authors declare: "Chinese college students are all over 18 years old and have gradually become independent and autonomous both psychologically and physically. They are influenced by their parents in the early stage and form a habitus of physical activity. After they step into the college campus, whether the influence on physical activity of parents sustain with previous teenage period needs to be tested. When teenagers grow up and get enrolled into college, the dominant groups they interacting with transits into their peer partners, such as friends, classmates and their roommates. In this context, individuals’ participation of physical activity may mainly be influenced by their peer partners rather tan parents."
What studies are based on to make this statement?
DISCUSSION
Is correct.
CONCLUSION
The conclusions should describe only the findings of the study carried out by the authors, not what was found in other studies. The results of other studies should only be mentioned in the discusión.
I believe that the authors exaggerate when making the following statement: "Our research can not only inform the Chinese government and education departments to formulate policies related to physical activity for Chinese college students, but also lead to better understanding of the impact of social support on physical activity".
His study is not based on government policies, nor on the relationship of government policies with the practice of physical activity.
In any case, the university is the one that must carry out these types of policies, strategies, offer different physical activity programs on the university campus.
Author Response
Dear Reviewer,
On behalf of my co-authors, we thank you very much for giving us an opportunity to revise our manuscript. We appreciate you very much for the positive and constructive comments and suggestions on our manuscript entitled "Physical Activity and Habitus: Parental Support or Peer Support?"(ijerph-2091561). In this revised version, we have addressed the concerns of your suggestions. An item-by-item response to the comments is enclosed, and the revision was marked in RED fonts in the manuscript. We hope that these revisions successfully address your concerns and requirements. Looking forward to hearing from you soon.
Sincerely,
Long Niu
Response to Reviewer 2 Comments
Point 1: I believe that the first paragraph should specifically specify the problem that the researchers are going to study. However, reading the first paragraph, it is not clear to me. They should emphasize the problem of physical inactivity in university students, The entire introductory section must be rewritten except for the last paragraph.
Response 1:Thank you very much for your suggestion. After careful consideration, we have rewritten the introduction part, which clearly emphasize the importance of physical exercise to physical fitness. At the same time, it also emphasizes the importance of strengthening physical exercise for Chinese college students according to the current situation that Chinese college students lack physical exercise.
Point 2: Page 3 Line 100/107. The authors declare: "Chinese college students are all over 18 years old and have gradually become independent and autonomous both psychologically and physically. They are influenced by their parents in the early stage and form a habitus of physical activity. After they step into the college campus, whether the influence on physical activity of parents sustain with previous teenage period needs to be tested. When teenagers grow up and get enrolled into college, the dominant groups they interacting with transits into their peer partners, such as friends, classmates and their roommates. In this context, individuals’ participation of physical activity may mainly be influenced by their peer partners rather tan parents."
What studies are based on to make this statement?
Response 2:Thank you very much for your suggestion. The problem may be that our expression is not accurate enough. What we mean here is that when Chinese college students enter college, the influence of parents' support on their physical exercise will decrease, while the influence of peer support will increase. We have revised it in the article.
DISCUSSION
Is correct.
CONCLUSION
Point 3: The conclusions should describe only the findings of the study carried out by the authors, not what was found in other studies. The results of other studies should only be mentioned in the discusión.
I believe that the authors exaggerate when making the following statement: "Our research can not only inform the Chinese government and education departments to formulate policies related to physical activity for Chinese college students, but also lead to better understanding of the impact of social support on physical activity".
His study is not based on government policies, nor on the relationship of government policies with the practice of physical activity.
In any case, the university is the one that must carry out these types of policies, strategies, offer different physical activity programs on the university campus.
Response 3:Thank you very much for your suggestion. We have deleted similar research conclusions in the conclusion part, and modified corresponding suggestions on government policies. It is emphasized that the research results of this paper can be used for future research on physical exercise of Chinese college students, so as to improve their physical health, as well as the real basis for realizing the strategic goal of healthy China as soon as possible.

Round 2
Reviewer 1 Report
Dear Authors,
Thank you for the effective revisions, the paper has been significantly improved.
Reviewer 2 Report
Its correct